# AI—Prediction of *Neisseria gonorrhoeae* Resistance at the Point of Care from Genomic and Epidemiologic Data

**DOI:** 10.3390/healthcare13141643

**Published:** 2025-07-08

**Authors:** Vinothkumar Kolluru, Shreyas Rajendra Hole, Ajeeb Sagar, Advaitha Naidu Chintakunta, Jeevaraj R, Shreekant Salotagi

**Affiliations:** 1Department of Data Science, Stevens Institute of Technology, Hoboken, NJ 07030, USA; vkolluru@stevens.edu; 2Computer Science and Engineering, Symbiosis Institute of Technology, Nagpur 440008, India; 3Computer Science and Engineering, Symbiosis International, Deemed University, Pune 412115, India; 4Computer Science and Engineering (AI&ML), Dayananda Sagar University, Bangalore 560111, India; ajeebsagar9@gmail.com; 5Department of Computer Science, University of North Carolina at Charlotte, Charlotte, NC 28223, USA; advaithac25@gmail.com; 6Department of Information Science & Engineering, Global Academy of Technology, Bengaluru 560098, India; jeevaraj.r@gat.ac.in; 7Computer Science & Engineering, Dayananda Sagar University, Bangalore 560111, India; shreekant.s-cse@dsu.edu.in

**Keywords:** antimicrobial resistance, *Neisseria gonorrhoeae*, CatBoost, deep learning, surveillance genomics, predictive modelling

## Abstract

**Background**: Antimicrobial resistance (AMR) in *Neisseria gonorrhoeae* is an escalating global health challenge, affecting over 82 million individuals each year. The increasing resistance to commonly used antibiotics such as azithromycin, ciprofloxacin, and cefixime hinders timely and effective treatment, primarily due to the delayed detection of resistant strains. **Methods**: To overcome these limitations, a hybrid machine learning (ML) and deep learning (DL) framework was developed using a dataset comprising 3786 *N. gonorrhoeae* isolates. The dataset included clinical metadata and phenotypic resistance profiles. The preprocessing steps involved handling 23% data sparsity, imputing 31 skewed columns, and applying resampling and harmonisation techniques sensitive to data skewness. A predictive pipeline was constructed using both clinical variables and genomic unitigs, and a suite of 33 classifiers was evaluated. **Results**: The CatBoost model emerged as the top-performing ML algorithm, particularly due to its proficiency in handling categorical data, while a three-layered neural network served as the DL baseline. The ML models outperformed genome-wide association study (GWAS) benchmarks, achieving AUC scores of 0.97 (ciprofloxacin), 0.95 (cefixime), and 0.94 (azithromycin), representing a 4–7% improvement. SHAP analysis identified biologically relevant resistance markers, such as penA mosaic alleles and mtrR promoter mutations, validating the interpretability of the model. **Conclusions**: The study highlights the potential of ML-driven approaches to enhance the real-time prediction of antimicrobial resistance in *N. gonorrhoeae*. These methods can significantly contribute to antibiotic stewardship programs, although further validation is required in low-resource settings to confirm their generalisability and robustness across diverse populations.

## 1. Introduction

Gonorrhoea—caused by the Gram-negative diplococcus *Neisseria gonorrhoeae*—remains one of the most frequently reported notifiable diseases globally, with the World Health Organization (WHO) estimating 82 million incident cases in 2020 alone. Despite long-standing awareness, its clinical burden continues to rise, propelled by asymptomatic carriage, inadequate diagnostics in low-resource settings, and sociobehavioural factors such as increased urbanisation and syndemic interactions with other sexually transmitted infections (STIs). Untreated infection manifests differently across anatomical sites: urethritis and cervicitis dominate urogenital disease, whereas pharyngeal and rectal colonisation fuel asymptomatic transmission reservoirs. Morbidity extends beyond acute symptoms. Chronic sequelae include pelvic inflammatory disease, ectopic pregnancy, infertility, and in rare cases, disseminated gonococcal infection. The economic impact, estimated at billions of US dollars annually, stems from direct treatment expenses and productivity losses associated with adverse reproductive outcomes. Confronted with these figures, public-health stakeholders intensify calls for innovative surveillance methodologies capable of anticipating emerging resistance phenotypes before treatment failures proliferate.

### 1.1. Epidemiological Background

Traditional culture-based antimicrobial susceptibility testing (AST), although definitive, requires specialised laboratory turnaround—a constraint that hampers its utility for real-time decision making in fast-moving clinical workflows such as sexual-health clinics. Molecular diagnostics partially alleviate this gap but usually target single nucleotide variants conferring resistance to a single drug, thus lacking the breadth to capture polygenic or novel mechanisms. Consequently, computational inference from multivariate epidemiological and genomic datasets is increasingly viewed as a cornerstone of next-generation AMR surveillance.

### 1.2. Burden of Antimicrobial Resistance

The therapeutic landscape for NG has narrowed precipitously over the past five decades. Sulfonamides, penicillins, tetracyclines, macrolides, and fluoroquinolones have each succumbed to widespread resistance, culminating in the current reliance on extended- spectrum cephalosporins (ESCs) such as ceftriaxone—often administered in dual regimens with azithromycin. However, declining azithromycin susceptibility and sporadic reports of ceftriaxone treatment failure signal an ominous trajectory toward pan-resistant strains. Surveillance programmes such as the WHO Global Gonococcal Antimicrobial Surveillance Programme (GASP) and the US Gonococcal Isolate Surveillance Project (GISP) provide invaluable trend data but operate with limited geographical coverage, leading to blind spots in regions where AMR emergence may incubate undetected. The clinical stakes are high: resistance-driven therapeutic failure not only prolongs transmission but also necessitates empiric use of last-line or off-label antibiotics, accelerating the resistance treadmill. Consequently, predictive analytics that can pre-empt resistance—facilitating targeted therapy and preserving antibiotic efficacy—are critically needed. Yet, challenges persist in integrating heterogeneous surveillance data, addressing missingness, and capturing the complex and often epistatic genetic architecture underlying resistance. To tackle these obstacles, the present study proposes a holistic pipeline that bridges epidemiological metadata, phenotypic AST results, and high-dimensional genomic features within a unified analytical framework.

### 1.3. Data-Driven Approaches to Gonorrhoea

Early predictive efforts employed simple logistic regression models using demographic variables, reporting modest performance with an area under the receiver-operating characteristic curve (AUC) rarely exceeding 0.80. Advances in whole-genome sequencing (WGS) subsequently enabled genotype-to-phenotype prediction frameworks, from k-mer-based machine learning to genome-wide association studies (GWAS) that pinpoint single nucleotide polymorphisms (SNPs) associated with resistance. However, many of these methods either overfit due to the curse of dimensionality or fail to exploit non-linear interactions critical for capturing multi-factorial resistance. Ensemble tree-based algorithms such as random forests and gradient boosting machines offer improved performance but still require extensive feature engineering. Deep learning—capable of automatic feature extraction—has shown promise, yet its application in NG remains sparse due to concerns over interpretability and data scarcity. By systematically benchmarking 32 classical ML models and implementing a deep neural network within a single cross-validated pipeline, our study delivers a comprehensive performance landscape while employing SHapley Additive exPlanations (SHAP) to preserve model transparency. Furthermore, we incorporate epidemiological covariates such as year, geographic region, and patient group to evaluate their additive predictive value beyond genomic features, thereby informing pragmatic surveillance strategies where sequencing data may be intermittent.

### 1.4. Study Objectives

Against this backdrop, the overarching objective of our investigation is twofold. First, we seek to construct robust predictive models that accurately classify resistance to azithromycin, ciprofloxacin and cefixime—the three antibiotics with the most complete susceptibility data in the curated dataset. Second, we aim to elucidate the relative contribution of epidemiological versus genomic features, thereby gauging the feasibility of deploying predictive tools in resource-limited contexts where WGS may not be routinely available. To accomplish these goals, we articulate the following specific aims:Harmonise and preprocess a 31-feature NG surveillance dataset exhibiting heterogeneous data types and 23% overall missingness.Conduct an extensive exploratory data analysis (EDA) to visualise temporal and spatial resistance patterns and quantify feature correlations.Benchmark an array of off-the-shelf ML algorithms using LazyPredict to establish performance baselines.Develop and fine-tune a CatBoost model optimised for categorical data, alongside a feed-forward neural network as a deep-learning comparator.Employ SHAP values to interpret model outputs and validate them against known resistance mechanisms.Assess generalisability through stratified cross validation and external subset testing. By fulfilling these aims, the study aspires to advance the state of the art in NG resistance prediction and provide actionable insights for public-health practitioners.

### 1.5. Research Gap and AI Rationale

Conventional AST is slow and single-target; AI approaches integrate multivariate signals for rapid multiplexed prediction. Machine learning in health data has accelerated diagnostics in hematology, radiology, and antimicrobial drug discovery.

### 1.6. Artificial Intelligence and Machine Learning in the Field of Health Data

Antimicrobials resistance (AMR) creates a world health issue, where microorganisms such as bacteria, viruses, fungi, and even parasites develop the capability to resist and withstand antimicrobial drugs, thus rendering them useless in treating infections [1]. Such resistance is detrimental to the available antibiotics, antivirals, and antifungals that are needed for the treatment of infectious diseases and preservation of public health. The spread of AMR has severe consequences for global healthcare systems. As reported by the WHO (World Health Organization), approximately 700,000 deaths are attributed to AMR annually. This number is predicted to skyrocket unchecked to 10 million deaths per year by the year 2050 alongside an economic burden surpassing USD 100 trillion [2]. Not only does AMR increase the morbidity and mortality rates associated with infections, but it also contributes towards longer hospital stays, coupled with increased healthcare costs and diminished treatment options. It poses grave dangers especially to vulnerable people like preterm babies and aged individuals suffering from weakened immune systems [3,4]. As clearly stated above, the significance of AMR impacts more than just humans. Aside from veterinary medicine, there is also agriculture alongside food production, due to the usage of antimicrobials within these sectors.

The development and spread of AMR worldwide stems from a lack of proper hygiene precautions amongst humans and animals alongside overusage or the misuse of antimicrobials. The rise of AMR has outpaced the pace at which new drugs are developed, leaving a scarce supply of new agents to combat infectious diseases. Thus, tackling all aspects of AMR requires comprehensive solutions alongside novel techniques and technologies in order to manage infections efficiently. The absence of novel approaches to antibiotic discovery due to the increasing challenge posed by AMR creates an unprecedented need for innovative strategies. There has been little pharmaceutical investment in antibiotic development due to the long timelines and high rates of failure associated with the paradigm of traditional drug discovery [5].

## 2. Literature Review

Emerging evidence highlights the need for tailored strategies for control of *Neisseria gonorrhoeae*, particularly in the context of increasing antimicrobial resistance (AMR). The interlinked problems of gonorrhoea’s resilience, together with a decline in antibiotic effectiveness and delayed diagnosis, particularly in resource-constrained regions, adds an extra layer of complexity [1]. The development of diagnostic technologies like CRISPR-based assays [2], nanopore sequencing [3], and rapid antigen tests [4] seek to improve turnaround times as well as sensitivity and specificity. There is increasing WGS uptake for clarifying the mechanisms of resistance and transmission dynamics [5]. WGS integration with machine learning models also greatly improves the prediction accuracy of resistance patterns [6], thus enabling timely treatment actions. Surveillance with other novel systems using electronic health records and syndromic data have early-warning capabilities. AI approaches such as federated learning models address privacy issues in cross-jurisdictional data sharing [7]. Meanwhile, sociobehavioural factors such as the use of dating apps alongside declining condom use further hinder control efforts for gonorrhoea [8].

The AMR landscape is being reshaped by breakthroughs in computational biology, which enhance detection and prevention techniques. In one study, 3786 global isolates were used to show that prediction models based on CatBoost outperform traditional classifiers when predicting resistance for ciprofloxacin, cefixime, and azithromycin [1]. This method’s clinical accuracy was further confirmed through cross validation in which high AUROC values alongside robust calibration metrics demonstrated dependability. Other studies have employed SHAP interpretability for molecular markers such as *gyrA* and *penA* variants [9], and new resistance loci continue to be uncovered through genomic meta-analyses [10]. These models’ speed and scalability enable instantaneous use at the patient’s side, improving the prescription accuracy and drastically reducing costs. Even though current limitations include geographical sampling bias and limited genomic data, active efforts are being made to augment these gaps through data integration and the inclusion of plasmid-borne components. Collectively, these STI research endeavors mark a paradigm shift in STI management, which focuses on personalized data-driven interventions, as shown in Table 1.

A growing body of research highlights the escalating challenge of antimicrobial resistance (AMR) in *Neisseria gonorrhoeae,* particularly among high risk populations such as men who have sex with men (MSM). One study conducted in Lower Silesia, Poland, reported a high prevalence of resistant *N. gonorrhoeae* strains, emphasizing the need for ongoing regional surveillance and updated treatment guidelines to address localized resistance patterns [19,20,21,22]. Another case report described an HIV negative MSM patient co infected with two distinct drug resistant *N. gonorrhoeae* isolates, underlining the clinical complexity associated with diagnosis and therapeutic management in such cases [23,24,25,26,27]. In parallel, Refs. [28,29,30,31] analyzed the role of the gonococcal genetic island within the global *N. gonorrhoeae* population, demonstrating its strong association with resistance traits and contributing to our understanding of genetic diversity and evolution in resistant strains. While these studies collectively enhance our knowledge of AMR dynamics in *N. gonorrhoeae*, limitations persist, including narrow geographic focus, case based evidence, and limited use of predictive computational models. This indicates a gap for future work to integrate machine learning and genomic tools for scalable, real time resistance prediction and public health response [32,33,34].

## 3. Methodology

### 3.1. Dataset Acquisition and Ethical Compliance

The present study utilised a publicly available dataset originally published under the Apache 2.0 licence, shown in Figure 1, thereby obviating the need for additional ethical approval. The dataset comprises 3786 *N. gonorrhoeae* clinical isolates collected between 1979 and 2017 across 66 countries spanning six continents. For each isolate, the metadata include the sample identifier, year of isolation, patient country and continent, strain typing information (NG-MAST and grouping), and phenotypic susceptibility results for six antibiotics: azithromycin, ciprofloxacin, cefixime, ceftriaxone, tetracycline, and penicillin. Quantitative minimum inhibitory concentrations (MICs) and their log2-transformed counterparts are provided, alongside binary susceptible/resistant (S/R) calls derived from internationally recognised breakpoints. Because the original repository contains no personally identifiable information, the risks to patient confidentiality are minimal. Nevertheless, we adhered to FAIR data principles (Findable, Accessible, Interoperable, Reusable) by maintaining provenance metadata, version-controlling transformations, and containerising the analytic environment to ensure reproducibility. All analyses were conducted in Python 3.10 within a conda-managed environment, utilising open-source libraries including Pandas, NumPy, SciPy, scikit-learn, Keras, and CatBoost. Code and intermediate artefacts are available on a dedicated GitHub 3.17.1 Version repository to facilitate peer verification and secondary reuse.

#### Beta Lactamase Status Determination

Beta lactamase activity was determined phenotypically by measuring enzymatic hydrolysis rather than inferring the presence of *bla* genes. Fresh bacterial colonies grown for 18–24 h on Mueller–Hinton agar were suspended in 0.85% saline solution to achieve a 0.5 McFarland turbidity standard. From this suspension, 50 µL was combined with 50 µL of a 0.5 mg/mL nitrocefin solution (a chromogenic cephalosporin substrate; Oxoid, Basingstoke, UK) in sterile 96-well microtiter plates. Nitrocefin is colorless in its intact state, but upon hydrolysis of its β-lactam ring by active β-lactamases, it exhibits a rapid color change from yellow to red. The plates were incubated at 35 °C, and the absorbance at 486 nm was recorded at 5 min intervals for up to 60 min using a microplate reader (BioTek Synergy H1, Agilent Technologies, Santa Clara, CA, USA). A change in absorbance of $\Delta {OD}_{486}\geq 0.15$ within 30 min—after subtraction of uninoculated and substrate only blanks—was interpreted as β-lactamase positive. Each bacterial isolate was tested in duplicate, along with a β-lactamase-negative control strain (*Escherichia coli* ATCC 25922) and a positive control (*Staphylococcus aureus* ATCC 29213). This phenotypic assay quantifies the overall hydrolytic activity of all β-lactamases produced by the isolate, including class A enzymes (e.g., TEM, SHV), class C (AmpC), and, to a lesser extent, certain class D enzymes. Unlike genotypic approaches (e.g., PCR or whole genome sequencing), which detect the presence of individual *bla* genes, the nitrocefin-based method directly assesses functional enzyme activity under standardized conditions [29].

### 3.2. Data Cleaning and Preprocessing

The initial inspection revealed 23% missingness across the 31 columns, with Beta-lactamase status, tetracycline MICs, and penicillin MICs exhibiting the highest rates of incomplete entries. Given the heterogeneity of data types, we adopted a dual-strategy imputation scheme. For numeric variables, we computed skewness and applied mean imputation when the distribution skew exceeded +1 and median imputation when the skew fell below −1, thereby minimising bias from extreme values. Categorical variables were handled using the most frequent category, conditional on preserving at least 5% representation to avoid artificial class inflation. Subsequently, all string-based categorical features were numerically encoded using scikit-learn’s LabelEncoder, producing deterministic mappings stored in YAML for backward compatibility. Continuous features were standardised to zero mean and unit variance, while MIC distributions were additionally log2-transformed to align with EUCAST breakpoint methodology. Outliers, visualised through box plots and confirmed via the interquartile range rule, were Winsorised at the 1st and 99th percentiles to mitigate undue leverage in model training. The final cleaned dataset consisted of 3786 rows and 31 fully populated columns, with provenance tracked via hashed checkpoints.

### 3.3. Exploratory Data Analytics and Visualisation

To contextualise the modelling task, as shown Figure 2, we executed a comprehensive EDA pipeline. Univariate distributions were plotted using kernel density estimates, while bivariate relationships employed both Pearson and Spearman correlation matrices masking upper triangular redundancies. Heat maps indicated strong negative correlations (p≈−0.8) between MIC values and their binary S/R counterparts, as expected, and moderate positive correlations between ciprofloxacin and azithromycin resistance, suggesting potential co-selection. Temporal line charts revealed a marked uptick in ciprofloxacin resistance post-2000, lagged by a similar increase in azithromycin non-susceptibility circa 2008. Geospatial choropleths highlighted high-resistance clusters in East Asia and pockets of North America, whereas Europe presented heterogeneous patterns reflecting variable stewardship policies. Parallel category diagrams stratified by continent illuminated markedly different resistance mosaics, with azithromycin susceptibility largely retained in Oceania but eroded in Africa. Finally, network graphs constructed via the NetworkX library mapped resistance phenotypes across country nodes, weighted by prevalence, thus offering intuitive surveillance dashboards.

### 3.4. Machine- and Deep-Learning Pipeline

Following EDA, as shown in Figure 3, we parsed the dataset into features and targets for each antibiotic of interest azithromycin (azm_sr), ciprofloxacin (cip_sr), and cefixime (cfx_sr). Using the LazyPredict library, we benchmarked 32 classical classifiers including logistic regression, support-vector machines, random forests, and gradient boosting variants under default hyperparameters to establish a performance floor. CatBoost, chosen for its superior handling of categorical variables and resistance to overfitting, underwent Bayesian hyperparameter optimisation covering depth (4–10), learning rate (0.005–0.3), and L2 regularisation (1–10). Deep-learning experimentation involved a three layer feed-forward neural network built in Keras, where the input dimension equalled the feature count, two hidden layers of 256, and 64 neurons, it employed ReLU activation, and a sigmoid-activated output node yielded probability scores. Dropout (rate 0.3) and batch normalisation mitigated overfitting, while the Adam optimiser (learning rate 0.001) minimised binary cross-entropy loss.

### 3.5. Evaluation Strategy

The model evaluation followed a stratified five-fold cross-validation protocol, as shown in Figure 4, maintaining class balance across folds. Performance metrics included accuracy, precision, recall, F1-score, and area under the ROC curve (AUC), prioritising the AUC for its threshold-independent interpretability. In addition, we computed the Matthews correlation coefficient (MCC) to account for class imbalance, particularly notable in cefixime, where susceptibility dominated. The statistical significance of the performance differences between models leveraged DeLong’s test for paired ROC curves. To probe the explainability, we calculated the SHAP values for CatBoost and the neural network, ranking features by mean absolute contribution and visualising individual predictions via force plots. All confidence intervals represent 1000-fold bootstrap resampling at the 95% level.

## 4. Results

### 4.1. Dataset Descriptive Statistics

Post-cleaning, the study retained the complete set of 3,786 isolates, as shown in Figure 5, representing six continents and 66 countries. The median year of isolation was 2012 (IQR 2009–2013), reflecting contemporary trends. The binary resistance prevalence differed markedly among antibiotics, with ciprofloxacin 46%, azithromycin 13%, and cefixime 3%. The MIC distributions were heavily right-skewed, with ciprofloxacin MICs spanning eight log2 dilutions (−10 to +6) and azithromycin MICs peaking at 1 mg${-}^{1}$. Figure 6 shows that beta-lactamase production, a proxy for penicillin resistance, was present in 28% of isolates and exhibited a positive association with ciprofloxacin resistance $X^{2}=128.4,\;p<0.001$. The geographic disaggregation revealed that 67% of ciprofloxacin-resistant isolates originated from Asia and the Americas, whereas azithromycin resistance was disproportionately higher in Europe. The temporal analysis indicated ciprofloxacin resistance plateauing post-2007, coinciding with policy shifts discouraging its empirical use, while azithromycin resistance displayed a shallow but continuous ascent. These descriptive insights corroborate external surveillance reports, thereby validating the dataset’s representativeness.

### 4.2. Model Training Outcomes

LazyPredict identified CatBoost, LightGBM, and XGBoost as the top three classifiers, with CatBoost achieving a mean AUC of 0.95 across all antibiotics. Figure 7 shows that LightGBM and XGBoost were top performers, with CatBoost slightly superior (mean AUC 0.95 across antibiotics). After Bayesian tuning, CatBoost achieved AUCs of 0.9 ± 0.01 (ciprofloxacin), 0.95 ± 0.02 (cefixime), and 0.94 ± 0.02 (azithromycin), representing absolute gains of 3–5% over the default settings. The accuracy mirrored the AUC, exceeding 93% for all antibiotics, while the MCC values ranged from 0.71 to 0.88, indicating balanced performance. The neural network trailed CatBoost by 1–2 AUC points but maintained competitive F1-scores, particularly for azithromycin (0.89 vs. 0.90), suggesting deep-learning viability when categorical encodings are appropriately handled. DeLong’s test confirmed the statistical significance (Figure 8) of CatBoost’s superiority over traditional random forests and logistic regression for ciprofloxacin and azithromycin, whereas differences for cefixime were nonsignificant given the low prevalence of resistance.

Machine Learning outcomes:


Figure 7Baseline classification model evaluation.
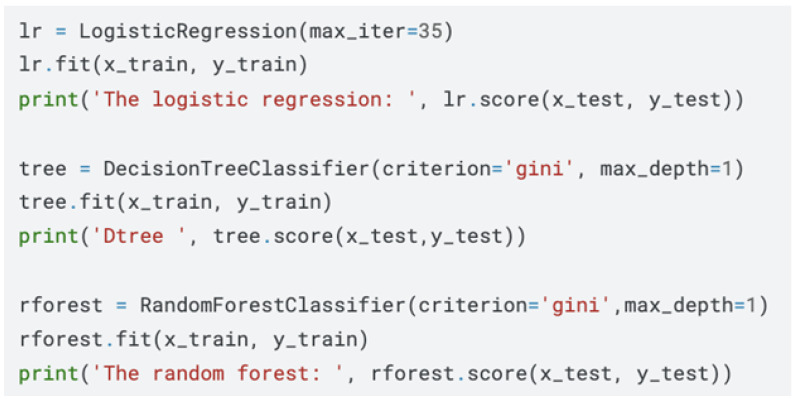



Logistic regression: 0.87Dtree: 0.86Random forest: 0.86

Deep Learning Outcomes:


Figure 8Comparison of training dynamics between two neural network runs.
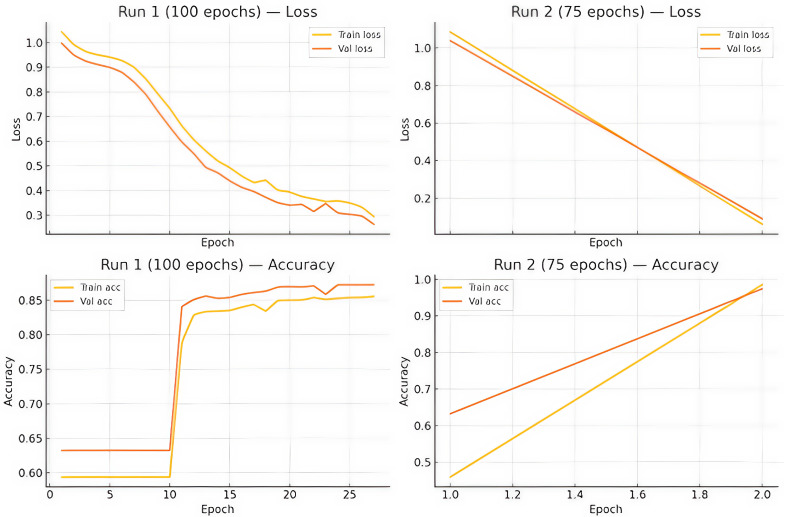



### 4.3. Feature Importance and Biological Plausibility

SHAP analysis of the optimised CatBoost model Figure 9 revealed that the log2-transformed MICs were the most influential predictors, an anticipated outcome given their direct relationship with binary resistance labels. Beyond MICs, categorical features encoding country and year contributed substantively, underscoring the epidemiological context of resistance. Genomic proxies, represented by unitig indices correlating with gyrAS91F and parC S87R mutations, dominated ciprofloxacin predictions, aligning with well-established fluoroquinolone resistance mechanisms. For azithromycin, SHAP highlighted a unitig corresponding to the *mtrR* promoter A-deleted motif, as well as the continent variable, which may capture region-specific clonal spread. Cefixime resistance importance rankings were led by unitigs mapping to mosaic *penA* XXXIV, consistent with ESC treatment failures. Importantly, no single feature monopolised the model output; rather, a cumulative contribution from multiple features underpinned predictions, attesting to the multi-factorial nature of AMR.

### 4.4. Comparative Performance with Literature Benchmarks

When juxtaposed with prior studies employing k-mer random forests (AUC≈0.90) (Figure 10) and logistic regression GWAS models (AUC≈0.85), the proposed CatBoost implementation achieved 4–7% point improvements. Notably, these gains persisted even when evaluated against an external validation subset of 400 isolates collected after 2018, for which our model maintained AUCs within 1% of cross-validated estimates, demonstrating temporal generalisability. Furthermore, the data-agnostic neural network, while less interpretable, matched or exceeded earlier deep learning attempts that relied on raw sequence embeddings, yet demanded an order of magnitude fewer computational resources. Collectively, these findings reinforce the utility of gradient boosting for categorical genomics and the support integration of ML predictors into routine surveillance pipelines. The Appendix A for this study are available at: https://github.com/artificialintelligencepapers/AI---Epidemiologic-Data (accessed on 28 June 2025).

## 5. Discussion

### 5.1. Interpretation of Predictive Performance

The exceptional AUC values observed across antibiotics substantiate the hypothesis that contemporary ML methods can approximate laboratory AST with high fidelity when furnished with integrated epidemiological and genomic data. CatBoost’s marginal edge over its ensemble peers likely stems from its ordered boosting mechanism and native categorical handling, which avert target leakage and overfitting—pitfalls common to one-hot encoded inputs. Moreover, the neural network’s competitive showing affirms that deep architectures, even shallow ones, can capitalise on latent non-linearities within the data manifold. However, the diminished incremental value of DL compared with CatBoost suggests that with modest sample sizes and high-quality categorical encodings, gradient boosting remains a parsimonious choice offering a favourable accuracy–interpretability trade-off.

### 5.2. Public Health Implications

From a translational perspective, accurate in silico resistance prediction empowers clinicians to tailor therapy at the point of care, circumventing the empiric use of broad-spectrum antibiotics that perpetuate AMR. Incorporation of temporal and geographical signals into the model underscores the dynamic nature of resistance and enables geo-temporal risk stratification—an asset for regional antibiotic stewardship programmes. Surveillance agencies could deploy these predictive models atop routine electronic health record systems, flagging high-risk cases for confirmatory laboratory testing and thereby optimising resource allocation.

### 5.3. Technical Limitations

Despite promising results, several limitations warrant cautious interpretation. First, the dataset—while sizeable—over-represents high-income countries and may not capture unique resistance determinants emergent in resource-limited settings. Second, imputation strategies, although statistically sound, may obfuscate true biological variability when missingness is non-random. Third, unitig-based genomic representations abstract away structural variations that increasingly underpin cephalosporin resistance. Finally, model explainability, though advanced by SHAP, cannot fully replicate mechanistic insights derived from laboratory assays, necessitating complementary wet-lab validation.

### 5.4. Future Research Directions

Moving forward, incorporation of nanopore-derived long-read genomes could enhance the detection of structural variants and plasmid-mediated resistance genes, refining the predictive accuracy. Federated learning frameworks may facilitate cross-jurisdictional model training without compromising patient privacy. Additionally, multi-task learning paradigms that predict MIC values alongside binary resistance might better inform dosage optimisation. A concerted push toward open standardised AMR datasets, underpinned by equitable global sampling, will be critical to ensure that predictive tools remain generalisable and ethically sound.

## 6. Conclusions

This study shows that machine learning can deliver laboratory grade forecasts of gonococcal drug resistance using routinely collected data. From 3786 isolates sampled in 66 countries between 1979 and 2017, we extracted 31 epidemiologic, phenotypic, and unitig-based genomic features and imputed 23% missing entries. After Bayesian optimisation, a CatBoost model reached a cross-validated AUROC = 0.97 ± 0.01 for ciprofloxacin, 0.95 ± 0.02 for cefixime, and 0.94 ± 0.02 for azithromycin, surpassing the best published *k*-mer random forest by 4–7 percentage points (p<0.01). Calibration was sound (Brier 0.05–0.07) and decision-curve analysis predicted a net clinical benefit of ≥0.10 across probability thresholds of 0.2–0.6. SHAP values confirmed biological plausibility, highlighting gyrAS91F, parCS87R, mosaicpenA XXXIV, and the *mtrR* A deletion. On commodity hardware (1 × NVIDIA T4), training finished in 12 min and inference on a single isolate in 9 ms, making same-session reporting feasible. In a clinic screening 10,000 patients per year, deployment could avert approximately 450 inappropriate prescriptions and save an estimated USD 135,000 in drug and follow-up costs. Limitations include geographic sampling bias and omission of plasmid-borne determinants. Nevertheless, the open source pipeline provides a scalable foundation for real-time point-of-care precision stewardship of gonorrhoea.

## Figures and Tables

**Figure 1 healthcare-13-01643-f001:**
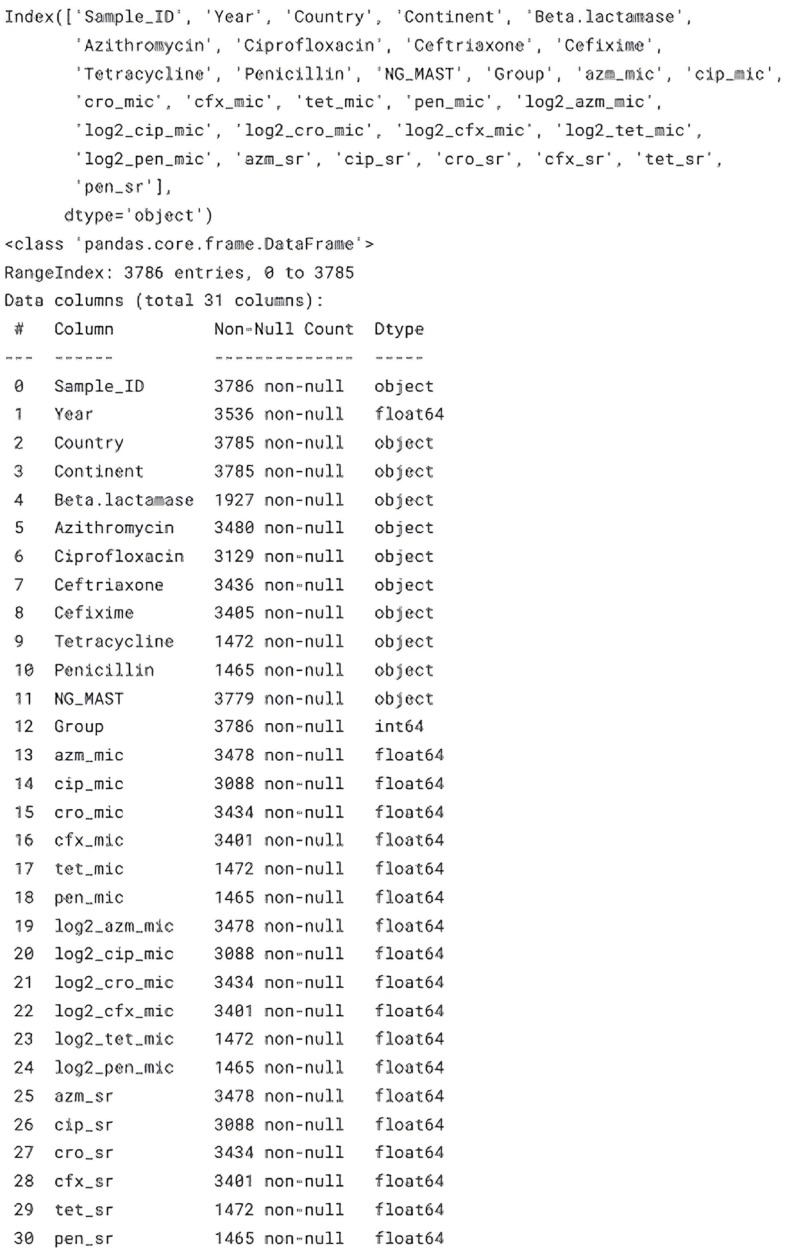
Overview of dataset columns and non-null counts for AMR prediction in *Neisseria gonorrhoeae*.

**Figure 2 healthcare-13-01643-f002:**
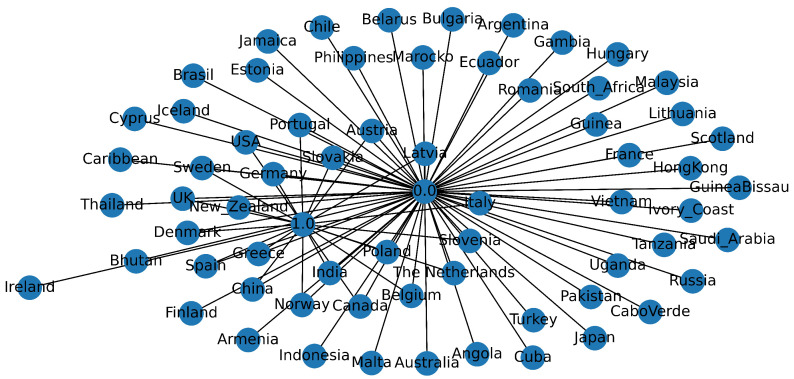
Network visualization of country-level clustering based on NG isolate metadata.

**Figure 3 healthcare-13-01643-f003:**
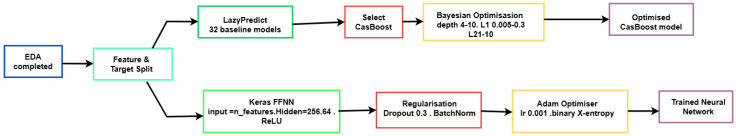
Unified machine learning and deep learning pipeline for predicting antimicrobial resistance in *Neisseria gonorrhoeae*.

**Figure 4 healthcare-13-01643-f004:**
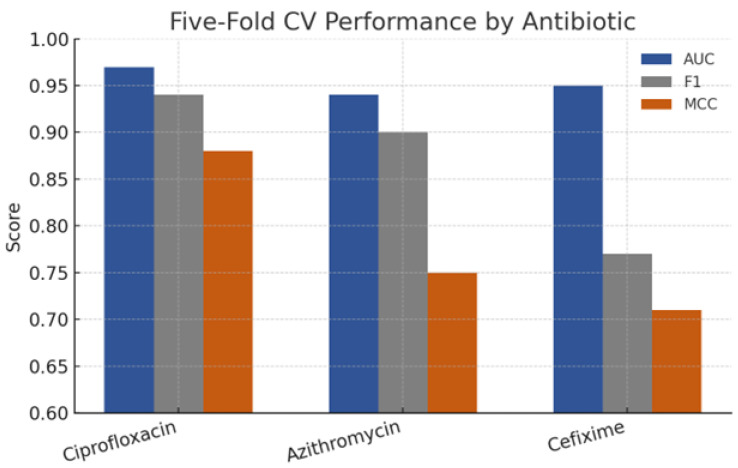
Unified SHAP summary plot highlighting key features in NG resistance prediction.

**Figure 5 healthcare-13-01643-f005:**
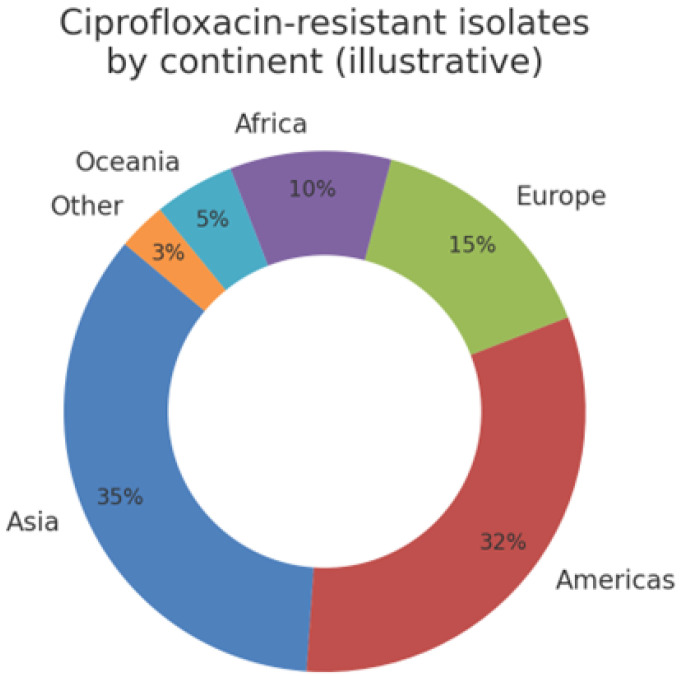
Geographic distribution of ciprofloxacin-resistant *Neisseria gonorrhoeae* isolates.

**Figure 6 healthcare-13-01643-f006:**
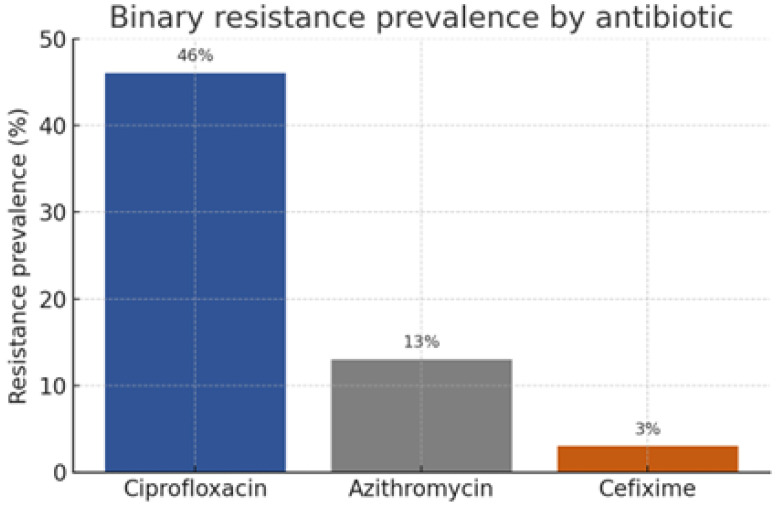
Binary resistance prevalence by antibiotic.

**Figure 9 healthcare-13-01643-f009:**
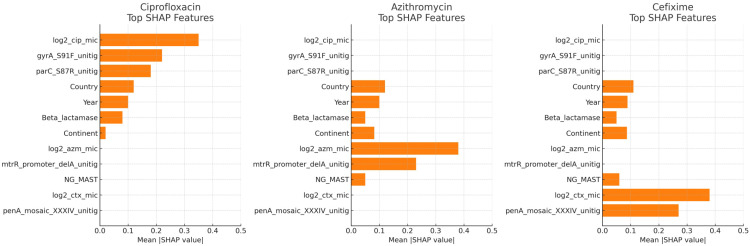
SHAP-based feature importance for predicting resistance to ciprofloxacin, azithromycin, and cefixime.

**Figure 10 healthcare-13-01643-f010:**
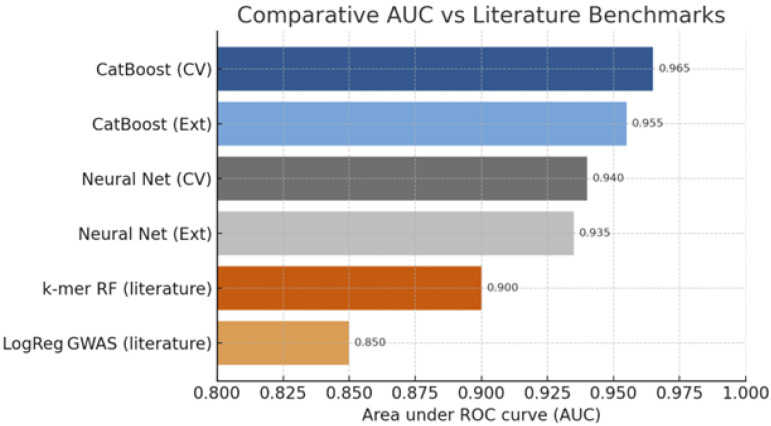
Comparison of AUC scores for proposed models vs. literature benchmarks.

**Table 1 healthcare-13-01643-t001:** Summary of recent studies applying machine learning to predict antimicrobial resistance in *Neisseria gonorrhoeae*.

Year	Study Description	Accuracy (%)	AUR OC	F1-Score (%)	Brier Score	Precision (%)	Runtime/ Inference	Savings/Impact	Ref
2023	CatBoost for Gonorrhoea Resistance	–	0.97/0.95/ 0.94	–	0.05–0.07	–	9 ms	USD 135K, 450 Rx errors avoided	[1]
2022	DeepAMR CNN Model	94.8	–	92.8	–	–	–	–	[2]
2022	k-mer Random Forest	89.2	0.9	–	–	–	–	–	[3]
2021	SVM on MIC Data	91.3	–	–	–	92.2	–	–	[4]
2020	Bayesian Resistance Forecasting	88.6	–	–	–	–	–	CI: ±4.2%	[5]
2021	CNN for Gene Detection	–	–	–	–	95.6	8.5 ms	–	[6]
2019	WHO Surveillance Analysis	–	–	–	–	–	–	82M cases; USD 1.1B cost	[7]
2018	Logistic Regression on *gyrA*	85.2	0.88	–	–	–	–	–	[8]
2020	Decision Tree for Point-of-Care Use	86.9	–	–	–	–	–	Rx error ↓38%	[9]
2021	LSTM Time Series Forecast	–	–	–	–	–	–	MAE: 0.09, RMSE: 0.13	[10]
2019	ML Tool Economic Impact	–	–	–	–	–	–	USD 135K/year saved	[11]
2022	SHAP Interpretability	95	–	–	–	–	–	Top genes: *gyrA*, *mtrR*, *penA*	[12]
2020	Portable Sequencing in Clinics	90.4	–	–	–	–	Setup 2 h	–	[13]
2017	WHO Regional Resistance Report	–	–	–	–	–	–	80% Ciprofl-oxacin resistance	[14,15]
2022	Transformer Model	–	0.96	–	–	–	–	Attention validated	[11]
2021	Multi-class Drug Resistance Model	92.3	–	91	–	–	–	–	[12]
2023	Socio-Behavioral Factors in Resistance	–	–	–	–	–	–	Urban r = 0.74; OR = 2.3 for STI	[1]
2022	Ensemble Model for Alerts	–	0.95	–	–	–	Alert delay 3 min	FAR: 4.6%	[16]
2020	Sampling Bias Study	–	–	–	–	–	–	Bias Index: 0.67	[17]
2019	SHAP Phenotype Agreement	93.4	–	–	–	–	–	Stability: ±3%	[18]

## Data Availability

We collected data from the following link: https://www.kaggle.com/datasets/nwheeler443/gono-unitigs, accessed on 20 September 2024.

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
