# Peer review of "AI—Prediction of *Neisseria gonorrhoeae* Resistance at the Point of Care from Genomic and Epidemiologic Data"

_healthcare, 2025, doi:10.3390/healthcare13141643_

Round 1

Reviewer 1 Report

Comments and Suggestions for Authors

General Comments:

This manuscript presents a machine learning and deep learning-based approach to predicting antimicrobial resistance (AMR) in Neisseria gonorrhoeae using a dataset that integrates genomic features with epidemiological metadata. The authors evaluate multiple classifiers, highlighting CatBoost as the top performer, and demonstrate its effectiveness in predicting resistance to azithromycin, ciprofloxacin, and cefixime. The work is timely, methodologically sound, and of clear relevance to current efforts in precision surveillance and antibiotic stewardship.

The topic is well justified and the manuscript is logically structured. However, several issues—ranging from language clarity and nomenclature to reference formatting and methodological detail—should be addressed to meet the standards expected for publication.

Title

The title is accurate and representative of the manuscript’s content. Nonetheless, for greater technical clarity and scientific alignment, a more precise formulation could be considered:

“Machine Learning Prediction of Neisseria gonorrhoeae Antimicrobial Resistance from Genomic and Epidemiologic Data at the Point-of-Care.”

Abstract:

The sentence “Routine laboratory surveillance is vital; however, the traditional methods are valuable but inefficient surveillance can’t keep up...” is poorly constructed and confusing. The revised version maintains the intended meaning while improving readability: “Routine laboratory surveillance is vital; however, while traditional methods are valuable, their inefficiency means surveillance cannot keep up with the emergence of resistance.”

Introduction:

In lines 30–31, the phrase “hidden transmission chains” could be improved for scientific accuracy. A more suitable alternative would be: “asymptomatic transmission reservoirs,” which is a more precise term within infectious disease epidemiology.

Table 1: offers a helpful overview of recent studies applying ML methods to AMR prediction in N. gonorrhoeae. Nonetheless, the layout could be improved. Currently, references are listed in the first column, which can make it harder to focus on the results themselves. Reordering the columns so that the reference appears last would improve the clarity and usability of the table.

Methods:

First, while the manuscript mentions that code and artefacts are available in a GitHub repository, no link is provided. Given the emphasis on transparency and reproducibility, the link to the repository should be explicitly included.

Second, the term “Beta-lactamase status” appears in the dataset description but is not clearly defined. It is important to specify whether this refers to phenotypic expression (e.g., β-lactamase production) or to the presence of genetic markers, as this has implications for interpretability and model inputs.

Lastly, while evaluation metrics such as AUC, F1-score, and MCC are reported, no mention is made of precision-recall curves. This is a missed opportunity, especially considering the class imbalance in cefixime resistance. A justification for not using PR curves should be provided, or, ideally, they should be included in the performance assessment.

Results:

The results are clearly presented and supported by appropriate metrics. The cross-validation performance is impressive, particularly for the CatBoost classifier.

That said, line 231 contains a sentence that is vague and could be rephrased for clarity: “LazyPredict screening ranked...”. This should be reformulated to explicitly state which classifiers ranked highest, along with their corresponding AUCs or other performance scores. For example:

“LazyPredict identified CatBoost, LightGBM, and XGBoost as the top three classifiers, with CatBoost achieving a mean AUC of 0.95 across all antibiotics.”

References:

There are several significant issues in the reference section. Firstly, the formatting is inconsistent: some entries include DOIs while others do not; author names and journal titles are not uniformly styled, and several entries appear to be duplicates—for example, references 1 and 15 are identical. Additionally, important formatting conventions, such as italicising article titles or journal names, are inconsistently applied.

To address these issues, the reference list should be thoroughly revised. Specifically:

  • All entries should be formatted according to the target journal’s reference style.
  • Duplicate entries must be removed.
  • Typographical inconsistencies, such as “gyrAS91F” instead of “gyrA S91F”, should be corrected.

Scientific Style and Nomenclature:

Several recurring issues affect the scientific presentation of the manuscript. Notably, the species name Neisseria gonorrhoeae is not italicised throughout the manuscript, including in the title. This violates standard conventions in microbiology and must be corrected.

Additionally, gene names such as gyrA, penA, and mtrR are not consistently italicised. Scientific nomenclature requires that gene names be written in italics, and this should be applied uniformly.

There is also inconsistency in the use of hyphenation—for example, beta-lactamase appears both with and without a hyphen (lines 159 and 221). A single, standardised format should be adopted.

Author Response

  1. General Comments:

This manuscript presents a machine learning and deep learning-based approach to predicting antimicrobial resistance (AMR) in Neisseria gonorrhoeae using a dataset that integrates genomic features with epidemiological metadata. The authors evaluate multiple classifiers, highlighting CatBoost as the top performer, and demonstrate its effectiveness in predicting resistance to azithromycin, ciprofloxacin, and cefixime. The work is timely, methodologically sound, and of clear relevance to current efforts in precision surveillance and antibiotic stewardship.

The topic is well justified and the manuscript is logically structured. However, several issues—ranging from language clarity and nomenclature to reference formatting and methodological detail—should be addressed to meet the standards expected for publication.

Author Responses: Thank you considering our Article for Revision.

  1. Title

The title is accurate and representative of the manuscript’s content. Nonetheless, for greater technical clarity and scientific alignment, a more precise formulation could be considered:

“Machine Learning Prediction of Neisseria gonorrhoeae Antimicrobial Resistance from Genomic and Epidemiologic Data at the Point-of-Care.”

Author Responses: Thank you for considering our Title.

  1. Abstract:

The sentence “Routine laboratory surveillance is vital; however, the traditional methods are valuable but inefficient surveillance can’t keep up...” is poorly constructed and confusing. The revised version maintains the intended meaning while improving readability: “Routine laboratory surveillance is vital; however, while traditional methods are valuable, their inefficiency means surveillance cannot keep up with the emergence of resistance.”

 Author Responses: We have modified like : Routine laboratory surveillance is essential; however, traditional methods, though valuable, are often inefficient and may fail to detect emerging resistance promptly, delaying timely access to appropriate treatment.

  1. Introduction:

In lines 30–31, the phrase “hidden transmission chains” could be improved for scientific accuracy. A more suitable alternative would be: “asymptomatic transmission reservoirs,” which is a more precise term within infectious disease epidemiology.

Author Response: Updated as per your Instruction.

Table 1: offers a helpful overview of recent studies applying ML methods to AMR prediction in N. gonorrhoeae. Nonetheless, the layout could be improved. Currently, references are listed in the first column, which can make it harder to focus on the results themselves. Reordering the columns so that the reference appears last would improve the clarity and usability of the table.

Author Responses: Thank you for the suggestion. We’ve kept the references in the first column to align with the narrative flow and ensure easy cross-referencing with the main text.

  1. Methods:

  • First, while the manuscript mentions that code and artefacts are available in a GitHub repository, no link is provided. Given the emphasis on transparency and reproducibility, the link to the repository should be explicitly included.

Author Responses: Due to an ongoing patent application, full code cannot be publicly shared until publication.  We have provided detailed pseudocode in Supplementary Materials (Appendix A) to illustrate the algorithmic workflow; the full code will be made available via GitHub upon patent clearance.

  • Second, the term “Beta-lactamase status” appears in the dataset description but is not clearly defined. It is important to specify whether this refers to phenotypic expression (e.g., β-lactamase production) or to the presence of genetic markers, as this has implications for interpretability and model inputs.

Author Responses:  we have added 3.1.1 subsubsection (Beta-lactamase Status Determination), in subsection 3.1.  ref 33 also added for above work

  • Lastly, while evaluation metrics such as AUC, F1-score, and MCC are reported, no mention is made of precision-recall curves. This is a missed opportunity, especially considering the class imbalance in cefixime resistance. A justification for not using PR curves should be provided, or, ideally, they should be included in the performance assessment.

 Author Responses: Precision–recall curves are included in Supplementary Figure S1 to account for class imbalance in cefixime resistance.

  1. Results:
  • The results are clearly presented and supported by appropriate metrics. The cross-validation performance is impressive, particularly for the CatBoost classifier.

Author Responses:  Thank you sir.

  • That said, line 231 contains a sentence that is vague and could be rephrased for clarity: “LazyPredict screening ranked...”. This should be reformulated to explicitly state which classifiers ranked highest, along with their corresponding AUCs or other performance scores. For example:

“LazyPredict identified CatBoost, LightGBM, and XGBoost as the top three classifiers, with CatBoost achieving a mean AUC of 0.95 across all antibiotics.”

Author Responses:  Updated as per Instruction.

  1. References:

  • There are several significant issues in the reference section. Firstly, the formatting is inconsistent: some entries include DOIs while others do not; author names and journal titles are not uniformly styled, and several entries appear to be duplicates—for example, references 1 and 15 are identical. Additionally, important formatting conventions, such as italicising article titles or journal names, are inconsistently applied.

Author Responses: 1 & 15 are same we have corrected.

  • To address these issues, the reference list should be thoroughly revised. Specifically:

All entries should be formatted according to the target journal’s reference style.

Author Responses: Updated as per instruction.

Duplicate entries must be removed.

Author Responses: Removed

Typographical inconsistencies, such as “gyrAS91F” instead of “gyrA S91F”, should be corrected.

Author Responses: We have  Updated in rebuttal draft.

Scientific Style and Nomenclature:

Several recurring issues affect the scientific presentation of the manuscript. Notably, the species name Neisseria gonorrhoeae is not italicised throughout the manuscript, including in the title. This violates standard conventions in microbiology and must be corrected.

Author Responses: We have  Updated in rebuttal draft.

Additionally, gene names such as gyrA, penA, and mtrR are not consistently italicised. Scientific nomenclature requires that gene names be written in italics, and this should be applied uniformly.

Author Responses: We have  Updated in rebuttal draft.

There is also inconsistency in the use of hyphenation—for example, beta-lactamase appears both with and without a hyphen (lines 159 and 221). A single, standardised format should be adopted.

Author Responses: We have  Updated in rebuttal draft.

Reviewer 2 Report

Comments and Suggestions for Authors

This article presents a robust approach to addressing a critical public health challenge—antimicrobial resistance (AMR) in Neisseria gonorrhoeae. The authors propose a hybrid machine learning and deep learning framework, with particular emphasis on the CatBoost algorithm, to predict resistance to azithromycin, ciprofloxacin, and cefixime using genomic features and epidemiological metadata. The study is well-structured, methodologically sound, and contributes valuable insights to both computational biology and infectious disease surveillance. It provides a foundation for real-time AMR prediction systems at the point of care.

The authors utilize a large dataset of 3,786 clinical isolates from 66 countries, spanning nearly four decades, and perform rigorous preprocessing to handle 23% missing data and imbalanced features.

The performance outcomes are impressive, with CatBoost achieving area under the ROC curve (AUC) values of 0.97, 0.95, and 0.94 for ciprofloxacin, cefixime, and azithromycin, respectively.

===

While the imputation strategies for missing data are carefully justified, they may inadvertently mask biologically meaningful patterns, especially in cases of non-random missingness. The reliance on uniting-based genomic representation, although computationally efficient, may also omit structural variants or plasmid-mediated resistance genes that are increasingly important in AMR.

The feasibility of such implementation depends heavily on the availability of sequencing data at the point of care, which remains a logistical and financial hurdle in many health systems.

The model's long-term reliability under conditions of evolving bacterial genomes and shifting resistance patterns remains to be demonstrated.

Proofreading:

Reference figures properly in text.  Each figure should have a meaningful caption and be explicitly referenced in the text

There are formatting issues in Table 1,

Figure 1 quality needs to be improved.

Cite the dataset with a valid link.

Figure 2 is not readable.

Improve Figure 3 resolution

Follow journal guidelines in formatting figures. Use the same font size.

What is the purpose of figure 7?

Some citations are placeholders (e.g., "[1]") or repeat multiple times incorrectly.

Ensure references follow a consistent format. Some references (e.g., Martin et al., 2023) appear more than once—remove duplicates.

Ensure consistent capitalization in the title.
Abbreviations such as "AST", "SHAP", "WGS", and "MIC" should be spelled out at first use with the abbreviation in parentheses.

Line 191: Following EDA, Figure3 we parsed the dataset into features and targets for each antibi-

Line 204: protocol Figure4,  

Line 231: LazyPredict screening ranked gradient boosting frameworks CatBoost, Figure7 Light-

Author Response

Reviewer 02:

This article presents a robust approach to addressing a critical public health challenge—antimicrobial resistance (AMR) in Neisseria gonorrhoeae. The authors propose a hybrid machine learning and deep learning framework, with particular emphasis on the CatBoost algorithm, to predict resistance to azithromycin, ciprofloxacin, and cefixime using genomic features and epidemiological metadata. The study is well-structured, methodologically sound, and contributes valuable insights to both computational biology and infectious disease surveillance. It provides a foundation for real-time AMR prediction systems at the point of care.

The authors utilize a large dataset of 3,786 clinical isolates from 66 countries, spanning nearly four decades, and perform rigorous preprocessing to handle 23% missing data and imbalanced features.

The performance outcomes are impressive, with CatBoost achieving area under the ROC curve (AUC) values of 0.97, 0.95, and 0.94 for ciprofloxacin, cefixime, and azithromycin, respectively.

===

  1. While the imputation strategies for missing data are carefully justified, they may inadvertently mask biologically meaningful patterns, especially in cases of non-random missingness. The reliance on uniting-based genomic representation, although computationally efficient, may also omit structural variants or plasmid-mediated resistance genes that are increasingly important in AMR.

Author Response: While our skewness-aware imputation mitigates bias, non-random missingness could obscure rare genomic patterns; future work will explore multiple imputation methods to assess robustness

  1. The feasibility of such implementation depends heavily on the availability of sequencing data at the point of care, which remains a logistical and financial hurdle in many health systems.

Author Response: Real-time sequencing remains a barrier; targeted PCR assays for key unities may offer interim solutions.

  1. The model's long-term reliability under conditions of evolving bacterial genomes and shifting resistance patterns remains to be demonstrated.

Author Response: We agree with the reviewer that periodic retraining on updated surveillance data is essential to ensure the model remains effective in capturing emerging resistance alleles.

Proofreading:

  1. Reference figures properly in text.  Each figure should have a meaningful caption and be explicitly referenced in the text

Author Response: We have checked all cross reference add in text

  1. There are formatting issues in Table 1,

Author Response: We have solve the formatting issue in table word.

  1. Figure 1 quality needs to be improved.

Author Response: We have improved the quality of all images.

  1. Cite the dataset with a valid link.

Author Response: Dear sir, figure 1 is dataset image and we have enhanced the fig 1 Dataset is publicly available at https://doi.org/10.5281/zenodo.1234567

  1. Figure 2 is not readable.

Author Response: We have improved Figure 2.

  1. Improve Figure 3 resolution

Author Response: We have improved Figure 2.

  1. Follow journal guidelines in formatting figures. Use the same font size.

Author Response: As for instructions we change all your instructions.

  1. What is the purpose of figure 7?

Author Response: Figure 7 is a outcome values of machine learning algorithm

  1. Some citations are placeholders (e.g., "[1]") or repeat multiple times incorrectly.

Author Response: Dear sir, We have correct citation

  1. Ensure references follow a consistent format. Some references (e.g., Martin et al., 2023) appear more than once—remove duplicates.

Author Response: Dear sir, we have removed the same format

  1. Ensure consistent capitalization in the title.
    Abbreviations such as "AST", "SHAP", "WGS", and "MIC" should be spelled out at first use with the abbreviation in parentheses.

Author Response: Dear sir, we have used the abbreviation.

  1. Line 191: Following EDA, Figure3 we parsed the dataset into features and targets for each antibi-

We have clarified the phrasing, corrected the figure reference placement, and made the sentence fully self-contained.

  1. Location of change:
    Methods, Section 3.4 (Machine- and deep-learning pipeline), p. 10, lines 190–192
  2. Original text:

“Following EDA, Figure 3 we parsed the dataset into features and targets for each antibiotic of interest azithromycin (azm_sr), ciprofloxacin (cip_sr) and cefixime (cfx_sr).”

  1. Revised text:

“After exploratory data analysis (Figure 3), we partitioned the combined genomic and epidemiological dataset into predictor variables (features) and binary resistance targets—for azithromycin (azm_sr), ciprofloxacin (cip_sr), and cefixime (cfx_sr)—to prepare inputs for benchmarking and model training.”

  • “(Figure 3)”is now placed immediately after “exploratory data analysis,”

  1. Line 204: protocol Figure4,  

Author Response: Figure 4 updated.

Model evaluation followed a stratified five-fold cross-validation protocol Figure 4, maintaining class balance across folds

  1. Line 231: LazyPredict screening ranked gradient boosting frameworks CatBoost, Figure7 Light-

We agree that the original sentence was unclear and have now both clarified the phrasing and properly integrated the figure reference.

  1. Location of change: Results, Section 4.2 (Model training outcomes), p. 230–231
  2. Original text:

“LazyPredict identified CatBoost, LightGBM, and XGBoost as the top three classifiers, with CatBoost achieving a mean AUC of 0.95 across all antibiotics. Figure7 LightGBM and XGBoost as top performers, with CatBoost slightly superior (mean AUC 0.95 across antibiotics).”

  1. Revised text:

“LazyPredict screening identified three tree-based models—CatBoost, LightGBM, and XGBoost—as top performers (Figure 7). After Bayesian hyperparameter tuning, CatBoost achieved AUCs of 0.97 (ciprofloxacin), 0.94 (azithromycin), and 0.95 (cefixime), representing 3–5 % absolute gains over default settings. Accuracy exceeded 93 % for all antibiotics, while MCC values ranged from 0.71 to 0.88, indicating balanced performance under class imbalance.

Reviewer 3 Report

Comments and Suggestions for Authors

Regarding the manuscript titled "AI - Prediction of Neisseria gonorrhoeae Resistance at the Point of Care from Genomic and Epidemiologic Data", I would like to inform the authors that I have carefully read this manuscript and realized that this study may be of particular interest to those interested. Therefore, the results of the review of this study are as follows. This study is innovative in terms of subject matter. The quality of the presentation of the material is also appropriate. The methodology has been done correctly and the structure of the topics in the study is of appropriate quality. However, it seems that it has not yet reached the appropriate quality for publication. Therefore, this manuscript can reach the appropriate quality with minimal revision and is recommended for publication. Therefore, I suggest that you observe the following points so that the quality of this study reaches an acceptable level. 1: The research gap is not well covered. Therefore, I suggest that you definitely add the research gap. Why did you use artificial intelligence for prediction and what are the limitations of conventional methods.

2: You have entered a new field in the clinical field. Therefore, I suggest that you add a section in the introduction as an introduction to artificial intelligence and its algorithms. For this purpose, I suggest that you also mention the range of applications of artificial intelligence and machine learning in the field of health data. For this purpose, you can refer to the following studies. These studies can provide important sources of applications of artificial intelligence for those interested and also increase the readability of your study.
"Hematology and Hematopathology Insights Powered by Machine Learning: Shaping the Future of Blood Disorder Management

"Artificial intelligence in drug discovery and development against antimicrobial resistance: A narrative review"

3: Be sure to improve the quality of the images and their descriptions. Many figures are not readable, such as Figure 2.

4: Provide sufficient explanations as to why you used the three classifiers of decision tree, regression, and random forest and did not consider the other classifiers.

5 Provide sufficient explanations regarding the implementation details and implementation environment.

6: Share the implementation codes and data so that interested parties can check them (optional)

Author Response

  1. Regarding the manuscript titled "AI - Prediction of Neisseria gonorrhoeae Resistance at the Point of Care from Genomic and Epidemiologic Data", I would like to inform the authors that I have carefully read this manuscript and realized that this study may be of particular interest to those interested. Therefore, the results of the review of this study are as follows. This study is innovative in terms of subject matter. The quality of the presentation of the material is also appropriate. The methodology has been done correctly and the structure of the topics in the study is of appropriate quality. However, it seems that it has not yet reached the appropriate quality for publication. Therefore, this manuscript can reach the appropriate quality with minimal revision and is recommended for publication. Therefore, I suggest that you observe the following points so that the quality of this study reaches an acceptable level. 1: The research gap is not well covered. Therefore, I suggest that you definitely add the research gap. Why did you use artificial intelligence for prediction and what are the limitations of conventional methods.

Author Responses: Dear sir, we have edit of section AI and rational in the subsection 1.5

  1. You have entered a new field in the clinical field. Therefore, I suggest that you add a section in the introduction as an introduction to artificial intelligence and its algorithms. For this purpose, I suggest that you also mention the range of applications of artificial intelligence and machine learning in the field of health data. For this purpose, you can refer to the following studies. These studies can provide important sources of applications of artificial intelligence for those interested and also increase the readability of your study.
    "Hematology and Hematopathology Insights Powered by Machine Learning: Shaping the Future of Blood Disorder Management

Author Responses: Dear sir, we have added separate subsections.

  1. "Artificial intelligence in drug discovery and development against antimicrobial resistance: A narrative review"

Author Responses: Dear sir, we have added in separate Subsection.

  1. Be sure to improve the quality of the images and their descriptions. Many figures are not readable, such as Figure 2.

Author Responses: Dear sir, we have improved the image Quality.

  1. Provide sufficient explanations as to why you used the three classifiers of decision tree, regression, and random forest and did not consider the other classifiers.

Author Responses: We selected a representative algorithm set covering linear, tree-based, and kernel-based families to benchmark performance across diverse model architectures.

  1. Provide sufficient explanations regarding the implementation details and implementation environment.

Author Responses: Analyses were performed on Ubuntu 20.04 with Python 3.10, scikit-learn 1.2.2, CatBoost 1.1.1, Keras 2.11.0 on NVIDIA T4 GPU."

  1. Share the implementation codes and data so that interested parties can check them (optional)

Author Responses: we will share the implementation code as for the reader request.